# LIPSCHITZ-AWARE LINEARITY GRAFTING FOR CERTI-FIED ROBUSTNESS

## ABSTRACT

Lipschitz constant is a fundamental property in certified robustness, as smaller values imply robustness to adversarial examples when a model is confident in its prediction. However, identifying the worst-case adversarial examples is known to be an NP-complete problem. Although over-approximation methods have shown success in neural network verification to address this challenge, reducing approximation errors remains a significant obstacle. Furthermore, these approximation errors hinder the ability to obtain tight local Lipschitz constants, which are crucial for certified robustness. Originally, grafting linearity into non-linear activation functions was proposed to reduce the number of unstable neurons, enabling scalable and complete verification. However, no prior theoretical analysis has explained how linearity grafting improves certified robustness. We instead consider linearity grafting primarily as a means of eliminating approximation errors rather than reducing the number of unstable neurons, since linear functions do not require relaxation. In this paper, we provide two theoretical contributions: 1) why linearity grafting improves certified robustness through the lens of the $l_\infty$ local Lipschitz constant, and 2) grafting linearity into non-linear activation functions, the dominant source of approximation errors, yields a tighter local Lipschitz constant. Based on these theoretical contributions, we propose a Lipschitz-aware linearity grafting method that removes dominant approximation errors, which are crucial for tightening the local Lipschitz constant, thereby improving certified robustness, even without certified training. After identifying dominant neurons based on an adversarially pre-trained model, we graft linearity into these neurons and fine-tune the model with existing adversarial training schemes. Our extensive experiments demonstrate that grafting linearity into these influential activations tightens the $l_\infty$ local Lipschitz constant and enhances certified robustness.

## 1 INTRODUCTION

The local Lipschitz constant is a fundamental property in neural network verification, as smaller values imply greater robustness to adversarial examples. It characterizes the network's sensitivity to input perturbations within a small region. The local Lipschitz constant has been widely used for certified robustness (Weng et al., 2018a; Jordan & Dimakis, 2020; Zhang et al., 2019b; Shi et al., 2022; Zhang et al., 2022; Huang et al., 2021). However, computing the exact Lipschitz constant is too costly, and even approximate estimates are often loose (Jordan & Dimakis, 2020). Moreover, identifying the worst-case adversarial examples Szegedy et al. (2014) is also known to be NP-complete (Katz et al., 2017). Consequently, despite its theoretical importance, effectively leveraging the local Lipschitz constant for improving certified robustness remains challenging.

On the one hand, relaxation-based verification approaches (Wong & Kolter, 2018; Raghunathan et al., 2018; Mirman et al., 2018; Zhang et al., 2019a; 2018; Weng et al., 2018a; Gowal et al., 2018), have been proposed to reduce computational complexity. However, approximation errors remain a significant obstacle for providing provable guarantees. These errors mainly arise when relaxing unstable ReLUs whose pre-activation intervals contain zero. To mitigate such errors, two major approaches have been broadly explored. The first is branch-and-bound (BaB) methods (Bunel et al., 2018; De Palma et al., 2021; Shi et al., 2025), which split unstable activation functions into sub-domains for complete verification. The second is certifiably robust training methods (Shi et al., 2021; De Palma et al., 2023; Lee et al., 2021; Mao et al., 2023), which incorporate bound infor-

mation, including unstable activations (Xiao et al., 2018; Shi et al., 2021; De Palma et al., 2022), into the training objectives. Ultimately, both verification and training approaches aim to reduce the approximation errors introduced by unstable ReLUs, which remain the primary bottleneck in tightening output bounds.

On the other hand, Linearity Grafting (LG) (Chen et al., 2022) is a prior method that directly replaces ReLUs likely to be both unstable and insignificant with linear functions, aiming to reduce the number of unstable ReLUs rather than stabilizing them. Consequently, LG successfully reduces the number of unstable neurons and enables scalable and complete verification, while improving certified robustness. However, it primarily focuses on identifying such neurons, and lacks theoretical insight into how this introduction leads to certified robustness improvements even without certified training. We instead consider linearity grafting primarily as a means of eliminating approximation errors rather than reducing the number of unstable neurons, since linear functions do not require relaxation.

Intuitively, reducing approximation errors narrows the gap between the upper and lower bounds of neural networks. This not only tightens the local Lipschitz constant but also improves certified robustness. Despite their impact on certified robustness, little attention has been paid to identifying and removing neurons that contribute significantly to approximation errors. Ideally, targeting ReLUs that have the greatest impact on the local Lipschitz constant would be most effective, especially when only a limited number of neurons can be modified.

Motivated by this intuition, we propose a Lipschitz-aware linearity grafting method that introduces linearity into ReLUs which produce dominant approximation errors —crucial for tightening the $l_\infty$ local Lipschitz constant— and thereby improves certified robustness.

Our contributions are summarized as follows:

- We theoretically analyze why grafting linearity into non-linear unstable ReLUs improves certified robustness through the lens of $l_\infty$ local Lipschitz constant. Since active and inactive ReLUs do not require relaxation, grafted ReLUs similarly bypass the need for relaxation.

- We demonstrate that grafting linearity into non-linear activation functions which are dominant source of approximation errors tightens the local Lipschitz constant further by considering the intervals between the upper and lower bounds of activation functions. Replacement of these activation functions with linear functions reduces the local Lipschitz constant and further improves certified robustness.

- We introduce a linearity grafting criterion, *weighted interval* score, to identify *influential* neurons with large weighted intervals that significantly affect the local Lipschitz constant in the next layer. Because the local Lipschitz constant measures the maximum sensitivity to input perturbation, reducing the contribution of such neurons is critical for tightening Lipschitz bounds. Grafting linearity into these neurons identified by the *weighted interval* score reduces the $l_\infty$ local Lipschitz constant to a level comparable to that of certifiably robust models.

- Additionally, we propose a novel *slope* loss, designed to stabilize unstable neurons by leveraging the slope of the upper bounds of ReLUs, and a *backward* neuron selection algorithm that considers the relationship between neurons in the consecutive layer in terms of the Lipschitz constant.

Our extensive experiments demonstrate that grafting linearity into influential ReLUs, identified by the *weighted interval* score, tightens the $l_\infty$ local Lipschitz constant and improves certified robustness.

## 2 RELATED WORKS

### 2.0 NOTATIONS

Let $f(\theta; x) : \mathbb{R}^{d_0} \to \mathbb{R}^{d_{L-1}}$, $W^{(\ell)} \in \mathbb{R}^{d_\ell} \times \mathbb{R}^{d_{\ell-1}}$, $b \in \mathbb{R}^{d_l}$, and $\sigma$ be a parameterized $L$-layer neural network, weight matrix, bias, activation function, respectively. Given an input $x \in \mathbb{R}^{d_0}$,

we define pre-activation values $z$, post-activation values $h$ as follows: $z^{(\ell)} = W^{(\ell)}h^{(\ell-1)} + b^{(\ell)}$, $h^{(\ell)} = \sigma(z^{(\ell)}) \ \forall \ell \in \{1, ..., L-1\}$ where $h^{(0)}(x) = x$. We refer a lower and upper bound of $z_i^{(\ell)}$, pre-activation values of $i$-th neurons in $\ell$-th layer, as $lb_i^{(\ell)}$ and $ub_i^{(\ell)}$.

## 2.1 RELAXATION OF NEURAL NETWORKS

Neural network verification ensures that a network satisfies given specification under all possible inputs within a defined perturbation. However, computing the exact worst-case adversarial examples is NP-complete (Katz et al., 2017). To mitigate this problem, various relaxation methods such as Interval Bound Propagation (IBP) (Gowal et al., 2018; Mirman et al., 2018), linear relaxation (Zhang et al., 2019a), semi-definite programming (Raghunathan et al., 2018) are proposed.

Linear relaxation methods approximate non-linear functions (e.g. ReLU) to certify neural networks by providing the linear bounds of neural networks:

$$A^L(x + \Delta) + b^L \leq f_i(x + \Delta) \leq A^U(x + \Delta) + b^U \tag{1}$$

where $A = W^L D^L W^{L-1} D^{L-1} \ldots D^1 W^1$, $W$ is weight matrices, $D$ is diagonal matrices of relaxed ReLUs (Zhang et al., 2019a). $D$ consists of upper and lower bounds of non-linear activation. In practice, neural networks verify whether the lower bound of the target class exceeds the upper bounds of all other classes for perturbed inputs. These lower bounds are computed by summing up upper and lower bounds from neurons in the previous layer, multiplied by negative and positive weights, respectively. This can be formulated as described below:

$$lb_i^{(\ell+1)} = \sigma\left(\sum_j w_{j,i}^- \cdot ub_j^{(\ell)} + w_{j,i}^+ \cdot lb_j^{(\ell)}\right) \tag{2}$$

This method is computationally efficient but incomplete, as it often produces "unknown" answer instead of "yes" or "no" answers. To achieve complete verification, branch-and-bound (BaB) method is used (Bunel et al., 2018; De Palma et al., 2021; Shi et al., 2025). Complete verifiers with BaB method split unstable neurons into sub-domains and proceed verification on these sub-domains recursively. However, the effectiveness of this complete verifier is heavily constrained by the number of unstable neurons, making verification infeasible for larger neural networks.

There also exists a line of work based on randomized smoothing (Cohen et al., 2019; Rekavandi et al., 2024) that certifies classification robustness by adding Gaussian noise to inputs, offering scalable probabilistic guarantees under perturbations. However, since the randomized smoothing methods do not explicitly address unstable neurons, we do not further investigate this line of work in this paper.

## 2.2 LIPSCHITZ CONSTANT FOR ROBUSTNESS

The Lipschitz constant $L$ of $f$ is defined as the smallest value such that for all $x, y \in \mathbb{R}^n$,

$$\|f(x) - f(y)\| \leq L\|x - y\|. \tag{3}$$

This constant measures the maximum rate of change of the function over the entire input space. However, computing the *global* Lipschitz constant—i.e., the supremum of the norm of the Jacobian over all inputs—is often computationally infeasible, and the resulting bound is typically too loose to capture meaningful behavior. In contrast, the local Lipschitz constant provides a tighter measure within a neighborhood of a given input, but computing it exactly is NP-complete and thus computationally intractable.

To overcome this, various works aim to compute tighter upper bounds for the local Lipschitz constant. CLEVER score (Weng et al., 2018b) uses random gradient sampling and extreme value theory to estimate local Lipschitz values. Other methods such as Fast-Lip (Weng et al., 2018a) and RecurJac (Zhang et al., 2019b) compute conservative Jacobian bounds through recursive layer-wise analysis. Furthermore, leveraging Jacobians with bound propagation and branch-and-bound refinement (Shi et al., 2022) achieves near-MIP-level tightness while it is scalable to relatively larger models.

Beyond analysis, the Lipschitz constant has been used in certifiably robust training (Huang et al., 2021; Zhang et al., 2022). Lipschitz-margin training, for example, directly incorporates bounds into

the loss function. Alternatively, Lipschitz-constrained architectures are designed to be 1-Lipschitz by construction using orthonormal weights or GroupSort activations. On the other hand, clipping upper bound of ReLUs with a trainable parameter tightens the local Lipschitz bound (Huang et al., 2021), while improving certified robustness.

In this paper, we focus on the $l_\infty$ *local* Lipschitz constant over $x$ in $\ell_\infty$-ball, $B(x_0, \epsilon) = \{x \in \mathbb{R}^n : \|x - x_0\|_\infty \leq \epsilon\}$. The local Lipschitz constant is then given by

$$Lip_\infty(f(x_0), \epsilon) = \sup_{\substack{x,x' \in B(x_0,\epsilon) \\ x \neq x'}} \frac{\|f(x) - f(x')\|_\infty}{\|x - x'\|_\infty}. \tag{4}$$

### 2.3 GRAFTING AND PRUNING FOR ROBUSTNESS

Pruning has been used to remove insignificant neurons or weights for model compression (Han et al., 2015; Liu et al., 2018). Network pruning is to remove neurons, themselves. On the other hand, Weight pruning can be categorized into unstructured pruning and structured pruning. Unstructured pruning, also known as weight pruning, cuts connections between neurons by zeroing out individual weights(Han et al., 2015). The unstructured pruning does not help reducing inference time or model size. In contrast, structured pruning removes more structured weights (e.g. filter, channel). With structured pruning, inference acceleration and model size reduction can be achieved (He et al., 2017). As the granularity of pruning increases, the model size can be reduced more significantly, but the information loss also becomes greater. To take advantages of the both filter pruning and channel pruning, clustering-based pruning methods (Zhong, 2022) are proposed. This pruning method not only improves inference time, but also remove model size simultaneously. Pruning neurons in a backward manner (Yu et al., 2018) is also studied that minimizes the reconstruction error of important responses based on final response layer.

From the perspective of removing insignificant neurons, pruning methods also used to improve robustness of neural networks(Wang et al., 2018; Sehwag et al., 2020; Ye et al., 2019; Vemparala et al., 2021; Liu et al., 2022). HYDRA (Sehwag et al., 2020) uses a gradient-based approach to iteratively identify and remove less important connections based on a robust loss with risk minimization, rather than relying on simple magnitude heuristics. HARP (Zhao & Wressnegger, 2024) learns per-layer pruning masks and rates to maximize robustness retention. By gradually increasing the sparsity and optimizing layer-specific pruning during fine-tuning, HARP achieves extreme compression (up to 99% parameter removal) with only minimal impact on adversarial accuracy. However, pruning for certified robustness has not been broadly explored (Sehwag et al., 2020; Lahav & Katz, 2021; Zhangheng et al., 2022; Zhao & Wressnegger, 2024).

Linearity grafting (Chen et al., 2022) is aligned with network pruning in that it identifies target neurons to control. This approach replaces them with linear functions with learnable slope and bias parameters. Especially, pruning can be viewed as a special case of linearity grafting when the slope and bias parameters are set to zero. However, it lacks theoretical insight into how this introduction leads to certified robustness improvement.

## 3 THE RELATIONSHIP BETWEEN GRAFTING LINEARITY AND $l_\infty$ LOCAL LIPSCHITZ CONSTANT

In this section, we demonstrate why linearity grafting improves certified robustness through the lens of the $l_\infty$ local Lipschitz constant, and show that grafting linearity into non-linear activation functions, the dominant source of approximation errors, yields a tighter local Lipschitz constant. It is worth noting that linearity grafting was originally proposed to reduce the number of unstable ReLUs without any theoretical insights.

**Lemma 1 (Approximation error of grafted ReLU)** *Let $ub$ and $lb$ be the upper bound and lower bound pre-actviation values of unstable ReLU. Then, the slope $\alpha^U$ and bias $\beta^U$ of the linear upper bound and the slope $\alpha^L$ and bias $\beta^L$ of the linear lower bound can be calculated as:*

$$\alpha^U = \frac{ub}{ub - lb}, \ \beta^U = -\frac{ub \cdot lb}{ub - lb}, \ \alpha^L = c, \ \beta^L = 0 \tag{5}$$

*where $0 < \alpha^L = c \leq 1$, $c$ is constant. Suppose $\gamma$ and $\omega$ be the slope and bias of a linear function of $x$, and $\alpha^L x + \beta^L \leq \gamma x + \omega \leq \alpha^U x + \beta^U$. Then, the ReLU's approximation error is greater than or equal to the grafted ReLU by the linear function, $\gamma x + \omega$.*

Lemma 1 demonstrates how grafting linearity into unstable ReLU tightens approximation errors. The proof of Lemma 1 is provided in Appendix D.

**Lemma 2 (Local Lipschitz constant of grafted network)** *Let $f : \mathbb{R}^{d_0} \to \mathbb{R}^{d_{L-1}}$ be a feedforward neural network, $x \in \mathbb{R}^{d_0}$ be an input, and $\epsilon$ be a perturbation budget. Suppose that we identify a set of unstable ReLUs for $\ell$-th layer, $\mathbb{U}^{(\ell)} = \{j \mid \mathrm{lb}_j^{(\ell)} < 0 < \mathrm{ub}_j^{(\ell)}\}$, and apply linearity grafting by replacing the ReLU of $j$-th neuron in $\ell$-th layer, $j \in \mathbb{U}^{(\ell)}$, with a linear function with slope $\gamma \leq 1$, then the $l_\infty$ local Lipschitz constant of the grafted network $f_{graft}$ satisfies:*

$$Lip_\infty(f_{graft}(x), \epsilon) \leq Lip_\infty(f(x), \epsilon). \tag{6}$$

Lemma 2 shows that grafting linearity into unstable ReLUs produces a tighter $\ell_\infty$ local Lipschitz constant when the slope of the grafted linear function is less than or equal to 1. The proof of Lemma 2 is provided in Appendix E.

**Theorem 1** *Let $s_i^{(\ell)} = \max_i(|w_{i,j}| \cdot |f_i^{U(\ell)} - f_i^{L(\ell)}|)$ be the score for $i$-th neuron in $\ell$-th layer, $\epsilon$ be a perturbation budget, and $\mathbb{G}_k \subset \mathbb{U}^{(\ell)}$ be the top-$k$ scoring neurons in terms of score $s$. Then, for any other subset $\mathbb{O}_k \subset \mathbb{U}^{(\ell)}$ of equal size not selected by score, and input $x$, grafting linearity into neurons in $\mathbb{G}_k$ leads to a tighter $l_\infty$ local Lipschitz bound:*

$$Lip_\infty(f_{graft}(x; \mathbb{G}_k), \epsilon) \leq Lip_\infty(f_{graft}(x; \mathbb{O}_k), \epsilon) \tag{7}$$

Theorem 1 demonstrates that applying linearity grafting with unstable ReLUs, which are the dominant source of approximation errors by the score function, tightens the $\ell_\infty$ local Lipschitz constant further than replacing randomly selected unstable ReLUs with linear functions. The proof is provided in Appendix F.

## 4 METHODS

Following the theorem, we propose a Lipschitz-aware linearity grafting method that introduces linearity into unstable yet influential ReLUs in a *backward* manner, aiming to reduce the local Lipschitz constants of neurons in the next layer. Additionally, we introduce a *slope loss* to stabilize unstable neurons by encouraging the slope of their upper bound to be close to zero or one.

### 4.1 NEURON SELECTION CRITERIA

We consider two criteria for neuron selection: weighted interval score $s_{wi}$, and instability score $s_u$.

**Weighted interval score.** Given a set of selected neurons $P^{(\ell+1)}$ in $\ell+1$ layer, the weighted interval score of $j$-th neuron in $\ell$-th layer, $s_{wi}^{(\ell)}(j)$, is defined as the maximum over an input set $\chi$ of the absolute value of intervals between the upper and lower bounds, multiplied by weights connected to the selected neurons only in the next layer. We refer neurons with high $s_{wi}$ scores as "**influential**" neurons to Lipschitz constant of neurons in the next layers in this paper.

$$s_{wi}^{(\ell)}(j) = \max_{i \in \chi} \max_{k \in P^{(\ell+1)}} |w_{j,k}^{(\ell+1)}| \cdot |ub_j^{(\ell)\{i\}} - lb_j^{(\ell)\{i\}}| \tag{8}$$

To tighten Lipschitz constant, we take into account the term $|w_{j,k}^{(\ell+1)}| \cdot |ub_j^{(\ell)} - lb_j^{(\ell)}|$ composing the calculation of Lipschitz constant. Especially, we presume that the upper and lower bound of the activation outputs are loosely their upper and lower bound values of pre-activations for the efficient calculation. Note that calculating the weighted interval score introduces negligible computational overhead, as it can be performed simultaneously with the instability score computation. All we need to do is solely keeps tracking minimum and maximum values of unstable neurons.

**Instability score.** The instability score of $j$-th neuron in $\ell$-th layer, $s_u(j, \ell)$, represents the number of inputs $\chi$ for which a neuron is unstable (Chen et al., 2022).

$$s_u(j, \ell) = \sum_{i \in \chi} \mathbb{1}[lb_j^{(\ell)\{i\}} < 0, ub_j^{(\ell)\{i\}} > 0] \tag{9}$$

where $\mathbb{1}$ is an indicator.

### 4.2 NEURON SELECTION METHOD

Since approximation errors propagate from previous layers, we identify neurons that are the most influential to the lower bounds of neurons in the next layer in a backward manner, Algorithm 1. By considering the connections of the selected neurons in the next layer, our method helps minimize the influence of these neurons. Neurons in $\ell$-th layers, except the last layer, are selected based on the following criteria in a *backward* manner. In this work, we *layer-wisely* select the top $15\%$ of neurons based on $s_{wi}^{(\ell)}$ from within the $80\%$ most *globally* unstable neurons. If all neurons in the last layer are selected, we retain $70\%$ of them based on $s_u$ to maximize the effectiveness of our method. For the remaining neurons, we select those with the highest $s_u^{(\ell)}$ scores. This criteria prioritize influential and unstable neurons that contribute most to Lipschitz constant of neurons in the next layer, while neurons in the last layer are selected solely based on the instability score, $s_u$.

### 4.3 *slope* LOSS FUNCTIONS FOR UNSTABLE RELU

We propose *slope* loss that is designed to stabilize unstable neurons by leveraging the slopes of ReLUs upper bounds. The *slope* loss makes the slopes deviate from $\frac{1}{2}$, encouraging them to be close to $1$ or $0$ corresponding to slopes of active or inactive ReLUs, respectively. The *slope* loss is also applied to slopes of the grafted neurons, since the slopes of the grafted neurons work similarly to the slopes of upper bound.

$$loss_{slope} = 1 - tanh(k \times (1-s)^2) \tag{10}$$

where $k = 2$ in this work, $s = \frac{ub}{ub-lb}$ for the unstable ReLUs and $s = \gamma$ for the grafted linearity ($\gamma x + \omega$). It is worth noting that $\alpha$-CROWN (Xu et al., 2020b) adjusts the slopes of ReLU lower bounds. At first glance, this appears similar to our slope loss, since both methods utilize ReLU bounds. However, the key difference is that our slope loss leverages the upper bounds, whereas $\alpha$-CROWN relies on the lower bounds.

## 5 EXPERIMENTS

We train four different sizes of model: CNN-B (Dathathri et al., 2020), ConvBig (Mirman et al., 2018), ConvHuge (17M), and ResNet4B (Bak et al., 2021). ConvHuge has $17M$ parameters so it seldom produce "Out-Of-Memory" (OOM) error due to numerous unstable neurons. Since the number of OOM we encountered is negligible ($< 10$), we ignore the OOM. Our method is implemented based on Auto-LiRPA (Xu et al., 2020a). Training details are described in Appendix C.

**Datasets.** These models are trained on MNIST (Deng, 2012), SVHN (Netzer et al., 2011), and CIFAR-10 (Krizhevsky et al., 2009) with $\epsilon = \frac{2}{255}$ except for the MNIST ($\epsilon = 0.1$) under $l_\infty$ perturbation. Due to the high volume of computation, we set calibration datasets sampled from the training datasets only for the calculation of both scores based on the model size: $4000$ samples from MNIST dataset for ConvBig, and CIFAR-10 dataset for CNN-B, $3000$ samples from SVHN dataset for ConvBig, CIFAR-10 dataset for ResNet4B and ConvBig, and $2000$ samples from CIFAR-10 for ConvHuge. Otherwise, we use the *FULL* training datasets to train models.

**Evaluation metrics.** We evaluate our method with five measurements: standard accuracy (SA %), robust accuracy (RA %), verified accuracy (VA %), unstable neuron ratio (UNR %), and verification time (Time, sec.). In more details, RA is measured by PGD-100 (Madry, 2017) with 100 restarts. UNR is the number of unstable neurons divided by the total number of neurons in the neural networks. Verification time is the amount of time to verify the datasets excluding misclassified or PGD-100 attacked. We set a wall time as 300 seconds, and $\alpha\beta$ CROWN (Zhang et al., 2018; Xu et al., 2020b; Wang et al., 2021) is used to measure VA and UNR. VA is evaluated with the first 1000 test datasets due to the high computational cost.

Table 1: We evaluate SA, RA, VA, UNR, and Time for neural networks trained with our masks and LG's masks. Ours outperform those trained with LG's mask.

| Method ($\epsilon = \frac{2}{255}$) | ConvBig, CIFAR-10 | | | | | CNN-B, CIFAR-10 | | | | | ConvHuge, CIFAR-10 | | | | |
|---|---|---|---|---|---|---|---|---|---|---|---|---|---|---|---|
| | SA | RA | VA | UNR | Time | SA | RA | VA | UNR | Time | SA | RA | VA | UNR | Time |
| Baseline | 86.76 | 73.98 | 1.5 | 17.75 | 121.61 | 80.13 | 63.00 | 36.70 | 16.74 | 133.64 | 89.56 | 74.06 | 0.20 | 17.32 | 159.80 |
| LG† (Chen et al., 2022) | 77.73 | 61.52 | 35.30 | 5.99 | 135.17 | 73.49 | 57.40 | 48.10 | 5.44 | 51.57 | 78.65 | 62.17 | 9.40 | 9.21 | 270.49 |
| Ours | 74.31 | 58.33 | **46.20** | 5.57 | 65.06 | 73.16 | 57.17 | **51.60** | 5.08 | 29.44 | 79.62 | 62.62 | **32.30** | 8.80 | 182.30 |

| Method ($\epsilon = \frac{2}{255}$) | ConvBig, MNIST $\epsilon = 0.1$ | | | | | ResNet4B, CIFAR-10 | | | | | ConvBig, SVHN | | | | |
|---|---|---|---|---|---|---|---|---|---|---|---|---|---|---|---|
| | SA | RA | VA | UNR | Time | SA | RA | VA | UNR | Time | SA | RA | VA | UNR | Time |
| Baseline | 99.19 | 97.39 | 92.10 | 17.78 | 15.77 | 77.80 | 60.17 | 0.50 | 20.44 | 34.00 | 88.76 | 73.88 | 13.30 | 13.47 | 213.17 |
| LG† (Chen et al., 2022) | 99.38 | 97.79 | **94.90** | 5.05 | 8.65 | 68.61 | 52.36 | 34.50 | 7.10 | 108.49 | 88.28 | 73.56 | 55.00 | 4.05 | 74.53 |
| Ours | 98.96 | 93.79 | 85.10 | 8.33 | 37.34 | 67.33 | 51.15 | **39.30** | 7.49 | 71.38 | 88.25 | 73.63 | **60.90** | 4.49 | 55.97 |

† stands for the reproduced results

Table 2: Our method with *slope* loss shows better VA compared to LG with $RS$ loss.

| METHOD ($\epsilon = \frac{2}{255}$) | ConvBig, CIFAR-10 | | | | | CNN-B, CIFAR-10 | | | | | ConvHuge, CIFAR-10 | | | | |
|---|---|---|---|---|---|---|---|---|---|---|---|---|---|---|---|
| | SA | RA | VA | UNR | TIME | SA | RA | VA | UNR | TIME | SA | RA | VA | UNR | TIME |
| LG w/ RS loss | 77.04 | 61.60 | 41.30 | 6.06 | 104.15 | 71.89 | 57.71 | 49.00 | 5.66 | 38.08 | 75.50 | 60.35 | 25.50 | 8.44 | 181.24 |
| Ours w/ slope loss | 69.39 | 55.88 | **50.60** | 2.94 | 26.50 | 71.09 | 56.44 | **52.00** | 3.80 | 14.27 | 68.86 | 55.80 | **51.40** | 1.21 | 31.25 |

| METHOD ($\epsilon = \frac{2}{255}$) | ConvBig, MNIST $\epsilon = 0.1$ | | | | | ResNet4B, CIFAR-10 | | | | | ConvBig, SVHN | | | | |
|---|---|---|---|---|---|---|---|---|---|---|---|---|---|---|---|
| | SA | RA | VA | UNR | TIME | SA | RA | VA | UNR | TIME | SA | RA | VA | UNR | TIME |
| LG w/ RS loss | 99.30 | 97.70 | **95.30** | 5.05 | 5.84 | 67.43 | 52.13 | 37.40 | 6.87 | 89.01 | 89.03 | 74.54 | 60.00 | 3.99 | 59.94 |
| Ours w/ slope loss | 97.65 | 86.65 | 81.80 | 0.25 | 14.34 | 65.30 | 50.73 | **43.60** | 6.16 | 45.99 | 88.34 | 74.68 | **68.00** | 1.61 | 25.86 |

## 5.1 Comparison of our method and LG

In this experiment, we evaluate our criteria without applying additional techniques such as $l_1$ regularization or small weight pruning. As shown in Table 1, ours outperform those trained with LG's masks in terms of VA, UNR, and Time, except for the experiment on the MNIST dataset. Specifically, our approach achieves VA improvements. However, the experiment on ConvBig-MNIST shows poor performances in VA, UNR, Time. Nevertheless, the average of SA and RA drops are only 0.75% and 1.22%, respectively. Overall, ours achieve superior verification performance while slightly reducing SA and RA.

## 5.2 Comparison of performances including *slope* loss and $RS$ loss

We evaluate the performance of our approach with the *slope* loss by comparing it to LG with the $RS$ loss. Additionally, we apply $l_1$ regularization and small weight pruning with a pruning ratio of 30% to introduce weight sparsity for further performance improvement. According to Table 2, our method with the *slope* loss outperforms LG with the $RS$ loss, except for the experiment on the MNIST dataset with the ConvBig model. While our method achieves better certified robustness, which is generally difficult to improve, LG obtains higher SA and RA. We attribute this to the difference in neuron selection: LG tends to graft neurons with small activation magnitudes, which correspond to low weighted-interval scores. Consequently, LG modifies neurons that have limited influence on sensitivity, resulting in better SA and RA but much weaker gains in certified robustness.

## 5.3 $l_\infty$ local Lipschitz constant of different training methods

As illustrated in Table 3, the CNN-B model trained with our method exhibits a tighter $l_\infty$ local Lipschitz constant compared to the model trained with adversarial training. In addition, Our Lipschitz constant obtained by our method is also smaller than that of the certifiably trained model, while still achieving higher VA. These results demonstrate that linearity grafting into influential ReLUs identified by our method not only tightens the local Lipschitz constant but also improves the certified robustness, thereby supporting the claim in Lemma 2.

Table 3: Comparison of other training methods w.r.t. $l_\infty$ Lipschitz constant of CNN-B models

| Training method | $l_\infty$ local Lipschitz constant | SA | VA |
|---|---|---|---|
| Adversarial training | 65.95 | 80.13 | 36.70 |
| Certifiably robust training | 20.82 | 61.47 | 49.50 |
| Ours w/ slope | 16.63 | 71.09 | 52.00 |

### 5.4 COMPARISION BETWEEN HIGHEST AND LOWEST WEIGHTED INTERVAL SCORES W.R.T. $l_\infty$ LOCAL LIPSCHITZ CONSTANT

As illustrated in the Table 4, linearity grafting with the highest weighted interval scores $s_{wi}$ results in a tighter $l_\infty$ local Lipschitz constant than the model with the lowest weighted interval scores. This result aligns with the arguments in Theorem 1: grafting linearity into ReLUs that have a dominant source of approximation errors leads to a tighter Lipschitz constant. In this work, the dominant source of approximation errors is measured by the weighted interval score.

Table 4: Comparison with grafting non-influential ReLUs w.r.t. $l_\infty$ Lipschitz constant of CNN-B models

| Grafting criteria | $l_\infty$ local Lipschitz constant | SA | VA |
|---|---|---|---|
| Lowest $s_{wi}$ scores | 17.49 | 71.87 | 51.70 |
| Highest $s_{wi}$ scores (Ours) | 16.63 | 71.09 | 52.00 |

### 5.5 COMPARISON BETWEEN *slope* LOSS AND *RS* LOSS

For a fair comparison between our *slope* loss and *RS* loss, we conduct an ablation study by combining our masking with *RS* loss and LG masking with *slope* loss. In this experiment, we do not train the slopes of grafted neurons to ensure fairness, but we still apply $l_1$ regularization and small weight pruning with 30% pruning ratio. As shown in Table 5, models trained with *slope* loss ②④ outperform those trained with *RS* loss ①③ w.r.t. VA, UNR, and Time. Comparing results (①-③ and ②-④), our criteria demonstrate better verification performances than LG's criteria. In terms of UNR, *slope* loss effectively stabilizes unstable neurons more than the *RS* loss, leading to a reduction in verification time.

Table 5: Comparison of criteria with *slope* loss and *RS* loss

| FAT ($\epsilon = \frac{2}{255}$) | ConvBig, CIFAR-10 | | | | |
|---|---|---|---|---|---|
| | SA | RA | VA | UNR | Time |
| ① LG w/ *RS* | 77.04 | 61.60 | 41.30 | 6.73 | 104.15 |
| ② LG w/ *slope* | 75.37 | 60.52 | 45.40 | 4.60 | 76.03 |
| ③ Ours w/ *RS* | 71.55 | 56.71 | 49.70 | 5.83 | 37.95 |
| ④ Ours w/ *slope* | 69.14 | 55.88 | 50.60 | 2.94 | 26.50 |

### 5.6 COMPARISON WITH CERTIFIABLY ROBUST TRAINING METHODS

We compare our method with other certifiably robust training methods. As shown in the Table 6, ours without *slope* loss outperforms IBP (Shi et al., 2021) w.r.t. SA, RA, and VA. Ours yields higher SA compared to other certifiably robust training methods. However, MTL-IBP (De Palma et al., 2023) and CTBench (Mao et al., 2024) methods show better robustness. Because certified training directly leverages the approximated bound on networks, it leads to SA drops. We discuss this trade-off between SA and VA more in Appendix B. However, our method solely with adversarial training requires dramatically less computational time than certified training methods. This reinforces the well-known fact that certified training is computationally expensive due to bound propagation and loosely approximated bounds in early training stages (Zhang et al., 2019a; Shi et al., 2021). Since all experiments in Table 6 use small network (2 conv + 2 MLP layers), we expect the training-time gap to widen further for larger models, emphasizing the scalability advantage of our method.

Table 6: Comparison with certifiably robust training methods on CNN-B, CIFAR-10.

| Method ($\epsilon = \frac{2}{255}$) | CNN-B, CIFAR-10 | | | | | |
|---|---|---|---|---|---|---|
| | SA | RA | VA | UNR | Time | Training Time (min.) |
| IBP (Shi et al., 2021) | 61.47 | 51.78 | 49.50 | 1.45 | 4.45 | 77.26 |
| MTL-IBP (De Palma et al., 2023) | 71.52 | 62.01 | 57.00 | 7.43 | 17.32 | 183.45 |
| CTBench (Mao et al., 2024) | 70.99 | 61.06 | 58.30 | 5.71 | 6.86 | 184.47 |
| Ours w/o *slope* | 73.16 | 57.17 | 51.60 | 5.08 | 29.44 | **38.57** |

## 5.7 EXPERIMENT ON DIFFERENT $\epsilon$

Table 7 shows that ours with $\epsilon = \frac{8}{255}$ outperform in terms of VA regardless of *slope* loss. In previous experiments, both methods benefited from the linearity grafting approach, leading to improvements in certified robustness. However, as the perturbation budget increases, the gains from LG become minimal, whereas our method continues to yield substantial improvement.

Table 7: Experiment on different target $\epsilon$ ($\frac{8}{255}$)

| Method ($\epsilon = \frac{8}{255}$) | CNN-B, CIFAR-10 | | | | |
|---|---|---|---|---|---|
| | SA | RA | VA | UNR | Time |
| LG (literature) (Chen et al., 2022) | 58.87 | 31.34 | 4.70 | 12.35 | 257.59 |
| Ours w/o slope | 52.89 | 29.51 | 15.70 | 14.27 | 126.69 |
| Ours w/ slope | 49.81 | 29.62 | 21.40 | 3.71 | 56.85 |

## 5.8 APPLICABILITY TO NON-RELU ACTIVATIONS

We observed that applying our method (w/o *slope* loss) with non-ReLU activations improves verified accuracy on Sigmoid networks (+1.6%) and Tanh networks (+5.9%), without performing any hyperparameter tuning. Although SA and RA dropped by about 2% in both cases, these initial results suggest that our method has the potential to generalize beyond ReLU-based architectures. The experiments are conducted under the same adversarial training setup in this paper.

Table 8: Experiments on non-ReLU activations on CNN-B with CIFAR-10

| Method ($\epsilon = \frac{2}{255}$) | Sigmoid | | | | Tanh | | | |
|---|---|---|---|---|---|---|---|---|
| | SA | RA | VA | Time | SA | RA | VA | Time |
| Adversarial training | 62.39 | 50.82 | 41.5 | 1.32 | 65.59 | 52.15 | 40.2 | 10.29 |
| Ours | 60.38 | 48.06 | 43.1 | 2.96 | 63.1 | 50.43 | 46.1 | 5.24 |

## 6 CONCLUSION

In this paper, we provide two theoretical contributions, previously underexplored, for how linearity grafting improves certified robustness through the lens of the local Lipschitz constant and how grafting linearity into non-linear activation functions, which are the dominant source of approximation errors, helps tighten the Lipschitz constant. Building on these theoretical insights, we propose Lipschitz-aware linearity grafting with *weighted interval* score to identify influential neurons with the greatest influence on the local Lipschitz constant. We further introduce *slope loss* to stabilize unstable neurons showing the reduced UNR, and *backward* neuron selection algorithm that considers the relationship between neurons in consecutive layers and the local Lipschitz constant. Our experiments align well with our claims from both theoretical contributions. In addition, they confirm that our method reduces the $l_\infty$ local Lipschitz constant to a level comparable to that of certifiably robust models, while significantly improving certified robustness.

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

## A  THE USE OF LARGE LANGUAGE MODELS (LLMS)

We acknowledge that LLMs were employed to polish the writing of our paper, primarily to improve grammar and readability. In line with the ICLR 2026 policy on LLM usage, we emphasize that such an assistant was limited to linguistic refinement, while all substantive ideas, analyses, and experiments are solely the work of the authors with full responsibility for the content presented herein.

## B  LIMITATIONS

As shown in Table 6, ours do not always outperform verifiably trained models. The main difference is that our method does not leverage upper bounds of worst-case adversarial examples during training. In contrast, state-of-the-art approaches incorporate both adversarial and certified losses to improve both empirical and certified robustness. Nevertheless, our method can be applied to adversarially pre-trained models, which are empirically robust but not verifiably robust. It is worth noting that our method enables such models to achieve verifiable robustness using only adversarial training. We leave a more thorough integration of our method with certified adversarial training as an important direction for future work.

**Trade-off between SA and VA**. To enhance certified robustness, certified training often sacrifices model utility, leading to a drop in SA. In particular, when enforcing tighter Lipschitz constraints, neural networks lose their expressivity (Yang et al., 2020). This trade-off arises because reducing the Lipschitz constant suppresses sharp decision boundaries and restricts networks' capacity. This trade-off highlights that improving certified robustness often comes at the cost of the model's utility and generalization ability. Therefore, it is crucial to maintain an appropriate balance between the model's expressivity and its certified robustness.

## C  TRAINING SETTING

**Loss**. For training neural networks, we use Fast Adversarial Training (FAT) (Wong et al., 2020) with Gradient Alignment (GA) (Andriushchenko & Flammarion, 2020) as LG did (Chen et al., 2022). It is already been demonstrated that introducing weight sparsity via $l_1$ regularization, small weight pruning, and $RS$ loss, helps verification (Xiao et al., 2018) and this can be applied to grafting linearity into ReLUs (Chen et al., 2022). We take advantage of this, except for the $RS$ loss. Thus the total loss function is the followings:

$$loss = loss_{FAT} + \lambda \times R_{GA} + \beta \times loss_{slope} + \gamma \times l_1 reg. \tag{11}$$

where hyper parameters $\lambda = 0.2$, $\beta = 0.00005$, and $\gamma = 0.0001$.

**Training settings**. We conduct all experiments on a single GPU, NVIDIA-RTX A6000 with 48GB memory. Except the hyper parameters for $slope$ loss, $l_1$ regularizer, and small weight pruning, we use the same configuration in (Chen et al., 2022) including $0.2$ coefficient for GradAlign, SGD optimizer with $0.9$ momentum, and $5 \times 10^{-4}$ weight decay: $0.1$ learning rate which is reduced by a factor of ten at 100 and 150 epochs, 128 batch size, $0.001$ learning rate for the remaining parameters, and $0.01$ learning rate for the slopes and intercepts of grafted neurons. We set the initial slope and bias for the grafted neurons to $0.4$ and $0.0$ following (Chen et al., 2022).

## D  PROOF OF LEMMA 1

**Proof.**

$$\alpha^U ub + \beta^U \geq \gamma\, ub + \omega \tag{12}$$

$$-(\alpha^L lb + \beta^L) \geq -(\gamma\, lb + \omega) \tag{13}$$

By summing up the Eq. 12 and 13, we have

$$\alpha^U ub + \beta^U - (\alpha^L lb + \beta^L) \geq (\gamma\, ub + \omega) - (\gamma\, lb + \omega) = \gamma(ub - lb) \tag{14}$$

$$\text{approximation error by the ReLU} \geq \text{approximation error by the grafted linear function} \tag{15}$$

## E    PROOF OF LEMMA 2

**Proof.** Base case: $\ell = 0$,

$$x_0 - \epsilon \leq f^{(0)} = x_0 \leq x_0 + \epsilon \tag{16}$$

$$Lip_\infty(f^{(0)}(x_0), \epsilon) = \frac{|f^{U(0)} - f^{L(0)}|}{|(x_0 + \epsilon) - (x_0 - \epsilon)|} = \frac{2\epsilon}{2\epsilon} = 1 \tag{17}$$

$$Lip_\infty(f_{graft}^{(0)}(x_0), \epsilon) = \frac{|f_{graft}^{U(0)} - f_{graft}^{L(0)}|}{|(x_0 + \epsilon) - (x_0 - \epsilon)|} = \frac{2\epsilon}{2\epsilon} = 1 \tag{18}$$

$$Lip_\infty(f_{graft}^{(0)}, \epsilon) = Lip_\infty(f^{(0)}, \epsilon) \tag{19}$$

For $\ell = k \geq 1$, let $\sigma^U$ and $\sigma^L$ be the element-wise upper linear bound and lower linear bound of ReLUs.

By Lemma 1,

$$
\begin{aligned}
Lip_\infty(f_j^{(k+1)}, \epsilon) &= \max_j \frac{|f_j^{U(k+1)} - f_j^{L(k+1)}|}{2\epsilon} \\
&= \max_j \frac{\sigma_j^{U(\ell+1)}(\sum_i W_{i,j}^{+(k+1)} \cdot f_i^{U(k)} - \sum_i W_{i,j}^{-(k+1)} \cdot f_i^{L(k)})}{2\epsilon} \\
&\quad - \frac{\sigma_j^{L(\ell+1)}(\sum_i W_{i,j}^{-(k+1)} \cdot f_i^{U(k)} - \sum_i W_{i,j}^{+(k+1)} \cdot f_i^{L(k)})}{2\epsilon} \\
&\geq \max_j \frac{\gamma_j^{(\ell+1)}(\sum_i W_{i,j}^{+(k+1)} \cdot f_i^{U(k)} - \sum_i W_{i,j}^{-(k+1)} \cdot f_i^{L(k)})}{2\epsilon} \\
&\quad - \frac{\gamma_j^{(\ell+1)}(\sum_i W_{i,j}^{-(k+1)} \cdot f_i^{U(k)} - \sum_i W_{i,j}^{+(k+1)} \cdot f_i^{L(k)})}{2\epsilon} \\
&= \max_j \frac{\gamma_j^{(\ell+1)}(\sum_i |W_{i,j}^{(k+1)}| \cdot |f_i^{U(k)} - f_i^{L(k)}|)}{2\epsilon} \\
&= \max_j \frac{|f_{graft\,j}^{U(k+1)} - f_{graft\,j}^{L(k+1)}|}{2\epsilon} \\
&= Lip_\infty(f_{graft\,j}^{(k+1)}, \epsilon)
\end{aligned}
\tag{20}
$$

$$Lip_\infty(f_{graft}^{(k+1)}, \epsilon) \leq Lip_\infty(f^{(k+1)}, \epsilon) \tag{21}$$

## F    PROOF OF THEOREM 1

**Proof.**

$$
\begin{aligned}
Lip_\infty(f_{graft\,j}^{(\ell)}(x; \mathbb{G}_k^{(\ell-1)}), \epsilon) &= \frac{\sigma(\sum_{i \notin \mathbb{G}^{(\ell-1)}} |W_{i,j}^{(\ell)}| \cdot |f_i^{U(\ell-1)} - f_i^{L(\ell-1)}|)}{2\epsilon} \\
&\quad + \frac{\gamma_j^{(\ell)} \sum_{k \in \mathbb{G}^{(\ell-1)}} |W_{k,j}^{(\ell)}| \cdot |f_k^{U(\ell-1)} - f_k^{L(\ell-1)}|}{2\epsilon}
\end{aligned}
\tag{22}
$$

$$Lip_\infty(f_{graft\ j}^{(\ell)}(x; \mathbb{O}_k^{(\ell-1)}), \epsilon) = \frac{\sigma(\sum_{i \notin \mathbb{O}^{(\ell-1)}} |W_{i,j}^{(\ell)}| \cdot |f_i^{U(\ell-1)} - f_i^{L(\ell-1)}|)}{2\epsilon}$$
$$+ \frac{\gamma_j^{(\ell)} \sum_{k \in \mathbb{O}^{(\ell-1)}} |W_{k,j}^{(\ell)}| \cdot |f_k^{U(\ell-1)} - f_k^{L(\ell-1)}|}{2\epsilon} \quad (23)$$

Assume that $Lip_\infty(f_{graft\ j}^{(\ell)}(x; \mathbb{O}_k^{(\ell-1)}), \epsilon) < Lip_\infty(f_{graft\ j}^{(\ell)}(x; \mathbb{G}_k^{(\ell-1)}), \epsilon)$.

$$\sigma\Big(\sum_{i \notin \mathbb{O}^{(\ell-1)}} |W_{i,j}^{(\ell)}| \cdot |f_i^{U(\ell-1)} - f_i^{L(\ell-1)}|\Big) + \sum_{k \in \mathbb{O}^{(\ell-1)}} \gamma_k^{(\ell)} |W_{k,j}^{(\ell)}| \cdot |f_k^{U(\ell-1)} - f_k^{L(\ell-1)}|$$
$$< \sigma\Big(\sum_{i \notin \mathbb{G}^{(\ell-1)}} |W_{i,j}^{(\ell)}| \cdot |f_i^{U(\ell-1)} - f_i^{L(\ell-1)}|\Big) + \sum_{k \in \mathbb{G}^{(\ell-1)}} \gamma_k^{(\ell)} |W_{k,j}^{(\ell)}| \cdot |f_k^{U(\ell-1)} - f_k^{L(\ell-1)}| \quad (24)$$

$$\sum_{k \in \mathbb{O}^{(\ell-1)} \setminus \mathbb{G}^{(\ell-1)}} \gamma_k^{(\ell)} |W_{i,k}^{(\ell)}| \cdot |f_k^{U(\ell-1)} - f_k^{L(\ell-1)}| < \sum_{k \in \mathbb{G}^{(\ell-1)} \setminus \mathbb{O}^{(\ell-1)}} \gamma_k^{(\ell)} |W_{i,k}^{(\ell)}| \cdot |f_k^{U(\ell-1)} - f_k^{L(\ell-1)}|$$
$$(25)$$

Right-hand side of the summation consists of the smallest $N - k$ elements, whose summation is the smallest sum when $N$ is the total number of neurons. By this contradiction,

$$\text{Lip}_\infty(f_{\text{graft}}(x; \mathbb{G}_k), \epsilon) \leq \text{Lip}_\infty(f_{\text{graft}}(x; \mathbb{O}_k;), \epsilon) \quad (26)$$

## G  ADVERSARIAL TRAINING WITH LINEARITY GRAFTING

We apply our method to two different FGSM variants, FGSM-MEP (Jia et al., 2022) and FGSM-PCO (Wang et al., 2024) on CIFAR-10 with ConvBig. As shown in Table 9, both linearity grafting methods (Ours and LG) substantially improve certified robustness over the baseline. Ours achieves the best certified robustness, whereas LG provides better SA and RA. This result highlights the trade-off between restraining Lipschitz constant and preserving generalization ability. Note that "catastrophic overfitting" is not considered in this work as it is out of scope.

Table 9: Experiments on various adversarial training

| Method ($\epsilon = \frac{2}{255}$) | SA | RA | VA | Time |
|---|---|---|---|---|
| FGSM-MEP baseline | 80.58 | 69.74 | 2.60 | 259.47 |
| FGSM-MEP w/ LG | 73.87 | 62.52 | 44.70 | 83.30 |
| FGSM-MEP w/ Ours | 70.42 | 58.97 | 46.00 | 67.31 |
| FGSM-PCO baseline | 81.65 | 69.26 | 5.10 | 277.51 |
| FGSM-PCO w/ LG | 74.07 | 61.35 | 40.90 | 96.51 |
| FGSM-PCO w/ Ours | 66.82 | 53.57 | 43.20 | 65.28 |

## H  LARGE-SCALE DATASET

We conduct experiments on CIFAR-100 with ConvBig model. In Table 10, although there is a 10% drop in SA, the improvement in certified robustness is significant. Moreover, our method outperforms other methods across every metric. These results demonstrate that our approach remains effective and scalable even on larger datasets.

Table 10: Experiments on CIFAR-100

| Method ($\epsilon = \frac{2}{255}$) | SA | RA | VA | Time |
|---|---|---|---|---|
| baseline | 53.28 | 36.41 | 1.8 | 127.86 |
| LG | 42.27 | 31.29 | 22.2 | 85.87 |
| Ours | 43.5 | 32.11 | 25.5 | 42.95 |

# I   AVERAGE SENSITIVITY

Our approach identifies neurons with the highest sensitivity measured by the *weighted interval* score as influential. By targeting these influential neurons, our intention is to more effectively constrain the Lipschitz constant. To validate this, we compare our method against a variant that performs linearity grafting based on average sensitivity. The results in Table 11 show that choosing neurons with the largest sensitivity leads to a tighter Lipschitz constant and improved certified robustness.

Table 11: Experiments on average sensitivity

| Method ($\epsilon = \frac{2}{255}$) | SA | RA | VA | Time | $\ell_\infty$ local Lipsthiz constant |
|---|---|---|---|---|---|
| Average sensitivity | 73.99 | 56.64 | 50.2 | 40.37 | 23.60 |
| Largest sensitivity (Ours) | 71.09 | 56.44 | 52.00 | 14.27 | 16.63 |

# J   EXPERIMENTS ON VARIOUS HYPER-PARAMETERS

## J.1   SLOPE LOSS

We conduct additional hyper-parameter searches on the slope loss, as shown in Table 12. First, when adjusting the slope learning rate ($\beta$), we observe a trade-off between SA/RA and VA discussed in Appendix B. By tuning the learning rate of the slope loss, we are able to control this balance. Next, when varying the slope parameter ($k$), the performance has been improved when $k = 4$. However, for consistency, we fix both the slope learning rate $\beta$ and $k$ values.

Table 12: Experiments on various hyper-parameter settings for slope loss

| slope learning rate $\beta$ | slope $k$ | SA | RA | VA | Time |
|---|---|---|---|---|---|
| $1e^{-4}$ | 2 | 66.95 | 54.21 | 47.60 | 23.84 |
| $5e^{-5}$ | 2 | 69.39 | 55.88 | 50.60 | 26.50 |
| $1e^{-5}$ | 2 | 71.09 | 56.75 | 48.50 | 45.26 |
| $5e^{-5}$ | 2 | 69.39 | 55.88 | 50.60 | 26.50 |
| $5e^{-5}$ | 4 | 69.86 | 55.92 | 50.70 | 27.78 |
| $5e^{-5}$ | 6 | 69.49 | 55.96 | 48.80 | 39.96 |

## J.2   WEIGHTED INTERVAL AND INSTABILITY SCORE

We do additional experiments by varying the scoring criteria used to choose neurons. In Table 13, we observe cases where VA increased under different ratio combinations. However, for consistency across datasets, we adopt the 80 instability - 15 weighted interval score as a default setting.

Table 13: Experiments on various hyper-parameter settings for weighted interval and instability score

| instability (%) | weighted interval (%) | SA | RA | VA | Time |
|---|---|---|---|---|---|
| 80 | 10 | 69.29 | 55.37 | 51.30 | 19.78 |
| 80 | 15 | 69.39 | 55.88 | 50.60 | 26.50 |
| 80 | 20 | 69.66 | 55.58 | 51.70 | 22.45 |
| 70 | 15 | 69.19 | 55.78 | 51.3 | 26.4 |
| 80 | 15 | 69.39 | 55.88 | 50.60 | 26.50 |
| 90 | 15 | 69.53 | 55.93 | 51.70 | 25.71 |

## K    CALIBRATION DATASET

We conduct ablation study on the size of the calibration dataset used to compute the weighted interval score and the instability score. As shown in the Figure 1, SA, RA, and VA show consistent performances regardless of the size of the datasets.

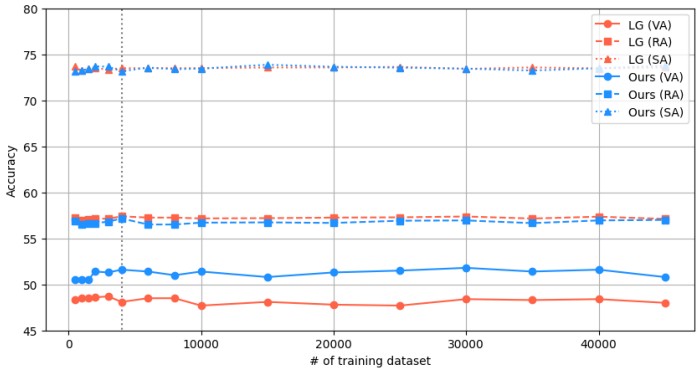

Figure 1: Comparison of SA, RA, and VA w.r.t. the size of training dataset used to compute the weighted interval score and instailibty score.

## L    VERIFICATION ON FULL TEST SET

We measure VA on FULL test set for our CNN-B model. Table 14 shows that VA of CNN-B model trained with our method decreases only $0.56\%$ for full test dataset. The results show that testing on 1000 test dataset is sufficient to measure the performance of the methods we used in experiments.

Table 14: Verification of Full test dataset on CNN-B, CIFAR-10.

| Size | SA | RA | VA |
|---|---|---|---|
| 1000 | 71.09 | 56.44 | 52 |
| 10000 | 71.09 | 56.44 | 51.44 |

## M    AVERAGE LOWER BOUND

As shown in the Figure 2, ours yield tighter average lower bounds compared to LG.

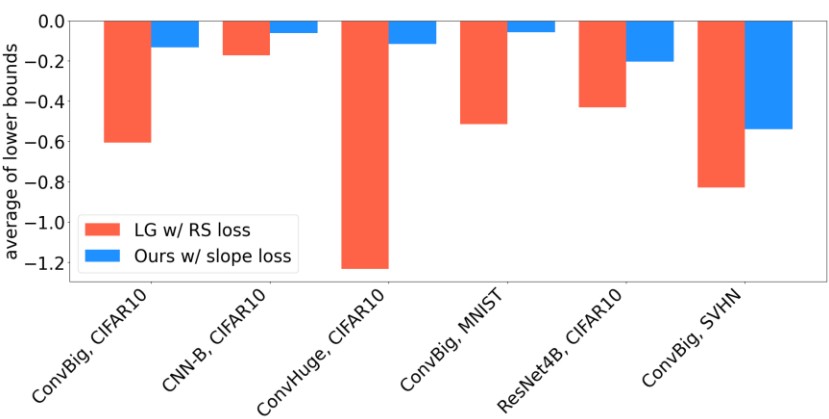

Figure 2: Average lower bounds of our models

## N  NEURON SELECTION ALGORITHM

---

**Algorithm 1** Neuron selection algorithm

---

**Input:** inputs $\chi$, weights $W$, instability score $s_u$, upper bound $ub$, lower bound $lb$, grafting ratio $r$

Init a set $\mathbb{G}$, weighted interval score $s_{wi}$

$\mathbb{G}^{(L-1)} \leftarrow$ Select 80% globally unstable neurons in $L-1$-th layer

**for** $\ell = L - 2$ **to** $0$ **do**

   **for** $j = 0$ **to** $J - 1$ **do**

      **for** $k = 0$ **to** $K - 1$ **do**

         $s_{wi}^{(\ell)}(j) = \max_{i \in \chi} \max_{k \in \mathbb{G}^{(\ell+1)}} |w_{j,k}^{(\ell+1)}| \cdot |ub_j^{(\ell)\{i\}} - lb_j^{(\ell)\{i\}}|$

      **end for**

   **end for**

   $\mathbb{G}^{(\ell)} \leftarrow$ Select 15% of influential neurons among 80% of unstable neurons

   $temp \leftarrow$ Select unstable neurons for the remainings

   $\mathbb{G}^{(\ell)} \leftarrow \mathbb{G}^{(\ell)} \cup temp$

**end for**

**return** $\mathbb{G}$

---

