# OpenReview forum: "Lipschitz-aware Linearity Grafting for Certified Robustness"
_ICLR.cc/2026/Conference — Submitted to ICLR 2026_

### Official Review · Reviewer_bFjd · 2025-10-17

**Soundness:** 2
**Presentation:** 2
**Contribution:** 3
**Rating:** 2
**Confidence:** 3

**Summary:**

The paper investigates how to improve local Lipschitz constants in ReLU neural networks.
This is done both theoretically as well as practical aspects are addressed.
The core components include identifying neurons using a weighted interval score and an instability score, as well as a slope loss encouraging unstable neurons to become more stable.

**Strengths:**

- The topic is of certifying robustness is important in safety-critical systems.
- The paper presents theoretical results on grafting linearity w.r.t. to $\ell_\infty$ local Lipschitz bound.
- Obtaining neural networks with fewer unstable neurons generally greatly improves verifiably due to a reduced accumulated approximation error.

**Weaknesses:**

- It appears that the approach is mainly about the training process; however, this only becomes clear later in the paper and is not adequately addressed, e.g., in the abstract.
- While Theorem 1 provides a nice theoretical result, a different scoring system is used in Sec. 4.
- The paper would benefit from more formalism in Sec. 4, e.g.,  how are lb/ub determined, what is $\chi$, ...
- No standard benchmarks, e.g., from latest VNN-COMP [1] are used.
- The comparison to other certified training approaches could be improved. See, e.g., [2] for a good overview.
- The presentation could be improved using some figures illustrating the process.

[1] Brix et al. "The fifth international verification of neural networks competition (vnn-comp 2024): Summary and results." arXiv. 2024.
[2] Koller et al. "Set-based training for neural network verification." TMLR. 2025.

Minor points:
- I think the reference in l.266 is broken (should reference Alg. 1?)

**Questions:**

- When unstable neurons are grafted and then verified, doesn't that mean that the original network remains unverified?
- Is the verified accuracy determined based on the computed Lipschitz bounds or is an external verifier used?
- Are the results solely on one evaluation run per method? If not, over how many runs are the presented training runs averaged? Can you also provide a standard deviation?
- How sensitive is your approach to the chosen hyperparameters (l.268-269, l.284, l.297)?

---

> ### Author Response · Authors · 2025-11-25
>
> We sincerely appreciate the reviewers for suggesting other benchmarks and providing careful reviews. Before addressing the questions, we would like to discuss the weakness.
>
> **W1**&**W6**. We add a brief description of our method in the abstract in the revised version. In addition, we plan to include a figure illustrating the overall process in the revised version.
>
> **W2**. As mentioned in Section 4.1, we presume that the upper and lower bounds of the activation outputs are approximated by the corresponding upper and lower bound values of the pre-activations for computational efficiency. Therefore, in Theorem 1, we treat $lb$ and $ub$ as equivalent to these upper and lower bound $f^U$ and $f^L$, respectively.
>
> **W3**. As stated in Section 2.0 Notation, $lb$ and $ub$ denote the lower and upper bounds of the pre-activation values. In addition, \chi referred to the input dataset, but for clarity, we have revised the notation to input set.
>
> **W4**. VNN-COMP is a competition designed to evaluate verifiers on a given network, rather than to certified training methods. For this reason, we did not adopt it as an evaluation benchmark in our work. Instead we add experiments on the CIFAR-100 to demonstrate that our method scales to larger datasets. We also incorporate additional baselines, including FGSM variants (FGSM-MEP and FGSM-PCO) and certified training method (CTBench), to further validate our method.
>
> [1] Jia, Xiaojun, et al. "Prior-guided adversarial initialization for fast adversarial training." ECCV 2022.
>
> [2] Wang, Zhaoxin, et al. "Preventing catastrophic overfitting in fast adversarial training: A bi-level optimization perspective." ECCV 2024.
>
> [3] Mao, Yuhao, Stefan Balauca, and Martin Vechev. "Ctbench: A library and benchmark for certified training." ICML 2025.
>
> W5. The method in [4] is implemented in MATLAB, and the authors do not provide the detailed training code used in their paper. This made it difficult for us to reproduce their results within the rebuttal period. As mentioned in our response to W4, we instead conducted additional experiments using CTBench, a certified training framework recently accepted to ICML 2025.
>
> [4] Koller, Lukas, Tobias Ladner, and Matthias Althoff. "Set-based training for neural network verification." TMLR 2025.
>
> **A1**. No, the original network remains unverified. Our approach simplifies the original network by modifying the network via linearity grafting, considering $\ell_\infty$ Lipschitz constant. Although the original model is not directly certified, verifying the grafted model yields significantly improved certified robustness. This outcome is generally hard to achieve without certified training.
> A2. The verified accuracy is evaluated using alpha-beta CROWN verifier[5]. The computation of $\ell_\infty$ Lipschitz constant is based on [6] with additional modification for linearity grafting.
>
> [5] Wang, Shiqi, et al. "Beta-crown: Efficient bound propagation with per-neuron split constraints for neural network robustness verification." NeurIPS 2021.
>
> [6] Shi, Zhouxing, et al. "Efficiently computing local lipschitz constants of neural networks via bound propagation." NeurIPS 2022.
>
> **A3**. We conduct all experiments only once following certified training literatures [3, 7]. We follow the standard practice in the certified training literature and conduct each experiment only once. However, to examine the consistency of our method, we additionally run the CNN-B model on the CIFAR-10 dataset with three different random seeds. As shown in the table below, the results exhibit stable SA, RA, and VA performance across runs, confirming the consistency of our approach.
>
> | seed | SA.   | RA    | VA    | Time  | UNR   |
> |------|-------|-------|-------|-------|--------|
> | 2    | 71.09 | 56.44 | 52.00 | 14.27 | 3.80  |
> | 3    | 70.66 | 56.17 | 51.70 | 23.54 | 12.14 |
> | 4    | 70.81 | 55.76 | 51.50 | 19.89 | 12.46 |
> | Avg. | 70.85 | 56.12 | 51.73 | 19.23 | 9.47  |
> | Std. | 0.22  | 0.34  | 0.25  | 4.67  | 4.91  |
>
> [7] De Palma, Alessandro, et al. "Expressive losses for verified robustness via convex combinations." ICLR 2024.
>
> **A4**. We conduct ablation studies on various hyperparameters and update them in the revised version.
> First, in Table 12 in the revised version, regarding the learning rate of the slope loss, we observe that decreasing the learning rate tends to increase SA and RA, while VA decreases accordingly. This suggests that the slope-learning-rate parameter can be used to control the balance between SA/RA and VA. Next, for the parameter $k$ performance remains stable up to $k=4$, but we observe a drop in VA when $k=6$. Lastly, in Table 13 in the revised version, we examine the effect of varying the ratio of neurons selected based on the instability score versus the weighted interval score. The results indicate that performance remains relatively stable near the setting of 80% instability and 20% influential neurons, showing no significant fluctuations around the ratio.

---

### Official Review · Reviewer_r39c · 2025-10-30

**Soundness:** 3
**Presentation:** 2
**Contribution:** 2
**Rating:** 4
**Confidence:** 4

**Summary:**

This paper investigates how linearity grafting—replacing parts of nonlinear activation functions with linear components—can enhance certified adversarial robustness by tightening local Lipschitz constants. The proposed Lipschitz-aware linearity grafting method aims to eliminate dominant approximation errors, thereby achieving tighter local Lipschitz bounds and improved certified robustness without requiring certified training.

**Strengths:**

The proposed approach improves upon the existing Linearity Grafting method and demonstrates potential in further tightening local Lipschitz constants.

**Weaknesses:**

* The authors state at the beginning of the abstract that “Lipschitz constant is a fundamental property in certified robustness, as smaller values imply robustness to adversarial examples.” However, despite obtaining a tighter local Lipschitz constant, the empirical results (Table 1) show a drop in robust accuracy (RA %). This seems contradictory to the intended goal of improving robustness. Could the authors clarify whether this behavior aligns with their theoretical claims?

* The standard accuracy (SA %) decreases significantly in most experiments, which undermines the contribution of this work. In extreme cases, one could simply remove a large proportion of unstable neurons to achieve a tighter Lipschitz bound but at the cost of severely degraded accuracy. The key challenge in Lipschitz-based methods lies in balancing SA and RA. The authors should discuss how their approach maintains this balance and why the accuracy degradation occurs.

* The proposed methods in Sections 4.2 and 4.3 are introduced rather abruptly and lack sufficient explanation. For instance, it is unclear how the specific proportions (e.g., 15 %, 80 %, 70 %) were determined (lines 268–270). Similarly, the design rationale for Eq. (5) and the choice of hyperparameters in Eq. (6) need to be clarified and justified.

* Table 7 presents results for a larger perturbation radius (ε), but the discussion is minimal. It would be helpful to elaborate on whether the proposed method can be easily extended to handle larger ε values and how this impacts performance.

* The empirical evaluation is relatively limited. It would strengthen the paper to include comparisons with more existing Lipschitz-based robustness methods, providing a broader context for the improvements claimed.

[1] Yang, Yao-Yuan, et al. "A closer look at accuracy vs. robustness." Advances in neural information processing systems 33 (2020): 8588-8601.
[2] Zhang, Bohang, et al. "Rethinking lipschitz neural networks and certified robustness: A boolean function perspective." Advances in neural information processing systems 35 (2022): 19398-19413.

**Questions:**

What is the formal definition of verified accuracy (VA %), and how does it relate to robust accuracy (RA %)? VA % seems to be introduced for the first time around line 321 without a clear explanation.

---

> ### Author Response · Authors · 2025-11-25
>
> We wholeheartedly thank the reviewer for the thoughtful comments and interest in the trade-off between SA/RA and VA. Before discussing the question, we would like to address the weaknesses.
>
> **W1**. Since our goal is to improve certified robustness (VA) rather than adversarial robustness (RA), drops in RA and SA are not unexpected. Although both adversarial and certified training optimize min-max objectives, they are slightly different. According to [1], adversarial training optimizes a lower bound on the inner optimization objective as adversarial examples cannot cover an entire input domain. In contrast, certified training optimizes an upper bound on the inner maximization objective, aiming to upper bound the worst-case adversarial examples over an entire input domain. However, these bounds are often loose in practice.
>
> From this perspective, certified training is designed to improve verified accuracy, even though this may come at the cost of robust accuracy, compared to adversarial training (e.g. FGSM variants, PGD-based training).
>
> [1] Mueller, Mark Niklas, et al. "Certified training: Small boxes are all you need." ICLR 2023.
>
> **W2**. We acknowledge that our method undermines SA, and we update a discussion of this limitation in Appendix B of the revised paper. In general, adversarial and certified training tend to smooth the decision boundary, which often leads to reduced utility performance. Nevertheless, given the difficulty of improving certified robustness, we believe that our main contribution lies in the improvement of VA. Furthermore, our additional studies suggest that appropriate tuning of the hyperparameters can achieve a desirable balance between SA/RA and VA, indicating the potential for mitigating this limitation.
>
> **W3**. We conduct extensive ablation studies on various hyper-parameters to validate our performance and the results are presented in Table 12 and 13 in Appendix J of the revised paper. First, when the learning rate of slope loss \beta\ decreases, SA and RA increase while VA decreases. We believe that this hyper-parameter plays a key role in balancing the trade-off between SA/RA and VA as we discussed in W3. We also find that overall performances remain consistent for values of $k \leq 4$. In addition, we observe that the method exhibits stable performance regardless of whether neurons are selected based on instability and weighted interval scores around 80% and 15%, respectively.
>
> **W4**. Ours consistently outperforms LG, regardless of whether slope loss is used. In other experiments, both methods benefited from the linearity grafting approach, leading to better certified robustness. However, as the perturbation budget increases, the gains from LG become minimal, whereas ours continues to yield substantial improvement (from 11% to 16%). This demonstrates that our approach is significantly more effective than LG in enhancing certified robustness, particularly under larger perturbation budgets.
>
> **W5**. We were unable to run an experiment for [2] due to dependency issues related to nvcc when executing the provided code. Since the method in [2] is based on Lipschitz-based training, we believe that including its results would have contributed to providing a broader context for the improvements claimed in our work. However, the issues made it difficult for us to reproduce their results within the rebuttal period.
>
> As an alternative, we incorporate experiments using CTBench [3], a certified training method. Similar to MTL-IBP, CTBench demonstrates improved VA but tends to decrease SA and RA. We assume that this tendency is caused from verified loss, which leverages an upper bound of the worst-case examples. In contrast, our method achieves better SA because it does not rely on a verified loss. We hope that this additional evaluation helps clarify the relative positioning of our method within the broader landscape of certified training.
>
> [2] Zhang, Bohang, et al. "Rethinking lipschitz neural networks and certified robustness: A boolean function perspective." NeurIPS 2022.
>
> [3] Mao, Yuhao, Stefan Balauca, and Martin Vechev. "Ctbench: A library and benchmark for certified training." ICML 2025.
>
> **A1**. Verified accuracy (VA) is the accuracy measured by a verifier such as alpha-beta CROWN [4] or Oval [5]. Since computing the worst-case adversarial example is known to be NP-hard, neural networks are bounded using linear relaxations under perturbations. However, these linear relaxations can be loose, which motivates the use of verifiers that iteratively split non-linear and unstable neurons into multiple linear cases to obtain more tighter bounds. In contrast, robust accuracy (RA) is measured by PGD attack with 100 restarts.
>
> [4] Wang, Shiqi, et al. "Beta-crown: Efficient bound propagation with per-neuron split constraints for neural network robustness verification." NeurIPS 2021.
>
> [5] Bunel, Rudy, et al. "Branch and bound for piecewise linear neural network verification." JMLR 2020.

---

### Official Review · Reviewer_bEmh · 2025-11-01

**Soundness:** 2
**Presentation:** 2
**Contribution:** 1
**Rating:** 4
**Confidence:** 3

**Summary:**

The main idea of this paper is to graft linearity into non-linear activation functions. This leads to lower approximate errors, make Lipschitz tight, and enhance certified robustness. In my onion, however, the paper still falls short of meeting the ICLR publication standard in its current form.

**Strengths:**

1.	The paper is well organized and clearly presented.
2.	The paper conducted extensive ablation studies to evaluate the effectiveness of the proposed method.

**Weaknesses:**

1.	The improvement is not significant and inconsistent. (as shown in Table 1 and 2)
2.	The core idea and methodology lack sufficient insight and novelty. Given that Lipschitz neural networks, such as LiResNet++ (Hu, 2024), have already scaled certified robustness to ImageNet and billion-parameter models, the proposed approach appears less competitive. Therefore, I would expect either a substantial performance improvement or a more conceptually innovative contribution.
[1] Hu, Kai, et al. "A Recipe for Improved Certifiable Robustness." ICLR, 2024.

**Questions:**

1.	My main concern is that the paper includes only one baseline. Although I have not reviewed every bound-propagation paper, it is difficult to believe that the most relevant baseline is from 2022. Could the authors include comparisons with more recent or stronger methods?
2.	In Table 3, the paper should also report the trade-off between clean and robust accuracy. While tightening (or suppressing) the Lipschitz constant is straightforward, it often comes at the cost of model utility. Demonstrating improvement beyond this trade-off would be crucial to show the true value of the method.
3.	The OOM statement on line 304 is unclear. Does it mean that the proposed method encounters out-of-memory (OOM) errors with some probability, and the training is restarted each time? Please clarify this behavior.
4.	The dataset used is relatively small. For more practical usage, can the paper consider cifar-100 or tiny-imagenet? Bound-propagation methods are often known to have scalability issues; providing detailed results on larger datasets would address this concern.
5.	For each attack evaluation, results are reported at only one perturbation budget. Providing results across multiple budgets would help readers assess performance trends. In addition, the chosen budgets are quite small. Please report robustness under larger budgets to evaluate the method’s behavior in more challenging regimes.

---

> ### Author Response · Authors · 2025-11-25
>
> We sincerely appreciate the reviewer for pointing out the importance of large scale dataset and other positive feedback! Before addressing the individual questions, we would like to clarify some of our contributions.
> - $\ell_\infty$ Lipschitz constant. Our focus is more on the $\ell_\infty$ Lipschitz constant rather than $\ell_2$ Lipschitz constant. As noted in [1], $\ell_2$ Lipschitz constant does not sufficiently capture locality. Although many prior works leverage $\ell_2$ Lipschitz constant during their training, the $\ell_\infty$ Lipschitz constant is rarely used due to its scalability issues. In particular, LG reports that computing $\ell_\infty$ Lipschitz constant requires several minutes per a SINGLE sample even for a small network (e.g. 2 conv + 1 mlp layers). This makes it infeasible to directly incorporate $\ell_\infty$ Lipschitz constant into training process, even though it provides much tighter local information.
> - Improving certified robustness with adversarial training for min-max optimization. In general, certified training optimizes the upper bound of max objective, whereas adversarial training optimizes lower bound on objective [2]. Recent certified training leverages the upper bound of outputs of the neural network by linear relaxation. However, the upper bounds are often loose and require substantial computational costs. However, adversarial training alone typically struggles to achieve meaningful certified robustness as shown in the Table 1, 2, and 9. Our work addresses this gap to enhance certified robustness without incurring the full cost of certified training.
>
> [1] Shi, Zhouxing, et al. "Efficiently computing local lipschitz constants of neural networks via bound propagation." NeurIPS 2022.
>
> [2] Mueller, Mark Niklas, et al. "Certified training: Small boxes are all you need." ICLR 2023.
>
>
> **A1**. We add two additional adversarial training baselines (FGSM-MEP [3] and FGSM-PCO [4]) and one certified training baseline (CTBench [5]) to further validate our claims. As shown in Table 9 in Appendix G of the revised paper, the linearity grafting approach significantly improves certified robustness, and our method achieves higher VA than LG.
>
> In Table 7, CTBench achieves higher VA than other certified training as well as Ours. However, since CTBench employs a verified loss, which optimizes an upper bound on the worst-case adversarial examples, we observe lower SA relative to Ours and MTL-IBP.
> Note that Ours is trained solely using adversarial training methods, which incur lower computational cost than certified training.
>
> [3] Jia, Xiaojun, et al. "Prior-guided adversarial initialization for fast adversarial training." ECCV 2022.
>
> [4] Wang, Zhaoxin, et al. "Preventing catastrophic overfitting in fast adversarial training: A bi-level optimization perspective." ECCV 2024.
>
> [5] Mao, Yuhao, Stefan Balauca, and Martin Vechev. "Ctbench: A library and benchmark for certified training." ICML 2025.
>
>
> **A2**. We update the manuscript and discuss this issue in Appendix B under Limitations. As expected, enforcing Lipschitz constant comes at the cost of model utility. This suppression leads to a drop in SA across our experiments in Table 1, 2, 6, 8, and 9. This trade-off arises because reducing the Lipschitz constant suppresses sharp decision boundaries and restricts networks' capacity. This trade-off highlights that improving certified robustness often comes at the cost of the model's utility. Therefore, it is crucial to maintain an appropriate balance between the model's expressivity and its certified robustness.
>
> **A3**. We would like to clarify that our method does not encounter any out-of-memory (OOM) issues. The OOM errors occurred only when calculating instability scores from the adversarially trained model (baseline) with 0.3% level. This indicates that the adversarially trained model may encounter OOM error during verification. This scalability issue can be addressed by linearity grafting, which is one of the main contributions of LG [2].
>
> **A4**. We apply our approach to CIFAR-100 ($\epsilon$=2/255) in Table 10. Our method achieves improved performance across SA, RA, and VA. Similar to the answer to Q1, linearity grafting (both Ours and LG) significantly improves certified robustness with adversarial training from 1.8% to 25.5%. Although our approach still exhibits a trade-off between clean and verified accuracy, these results demonstrate that our method remains effective on large-scaled dataset.
>
> **A5**. We already have experiments with a larger perturbation budget (8/255) in Table 7. While other experiments suggested that LG also improves VA, indicating that linearity grafting itself can be beneficial, we observe a different trend under a larger perturbation budget. Specifically, LG provides little to no improvement in VA, whereas our method consistently increases VA by at least 15%. This result highlights that our approach remains effective even when the perturbation budget becomes large.

---

### Official Review · Reviewer_SQiX · 2025-11-09

**Soundness:** 2
**Presentation:** 1
**Contribution:** 2
**Rating:** 2
**Confidence:** 4

**Summary:**

This paper investigate how different non-stable ReLU neurons affects the local Lipschitz constant of the model when we consider the adversarial perturbations. It proposes two metrics "weighted interval score" and "instability score" which enables us to select the unstable neurons more critical to model's Lipshcitz constant and thus verified robustness. By grafting such neurons by a linear function, we can improve the verified robustness of the model.

**Strengths:**

++ It is novel and interesting to consider verified robustness from the perspective of controlling Lipschitz constant and grafting.

++ The proposed method is generic and plug-and-play.

++ The intuition is theoretically justified to some degree.

**Weaknesses:**

1. The proposed framework introduces a lot of hyper-parameters, including the number of grafted neurons each layer, $k$ in Equation (5), $\lambda$, $\beta$ and $\gamma$ in Equation (6). I believe all these hyper-parameters will affect the performance to some degree. However, I did not see ablation studies or adequate discussions about them.

2. The experiments are weak in general. Comparisons with more robustness certification and the corresponding provable training algorithms should be included. For example, Table 6 considers MTL-IBP, its successors like alpha-beta-CROWN should be included.

3. Theoretical claims seem not rigorous. I have concerns in the proof of Lemma 1, it is unclear to have $|f_i^{U(k)} - f_i^{L(k)}| \geq |f_{graft, i}^{U(k)} - f_{graft, i}^{L(k)}|$ for **all values of $I$** in Equation (15). The induction assumption only indicates $max_i |f_i^{U(k)} - f_i^{L(k)}| \geq max_i |f_{graft, i}^{U(k)} - f_{graft, i}^{L(k)}|$, as indicated in Equation (11). However, I do not see why this can generalise to any value of $k$. The authors should provide more theoretical details about this.

4. The theoretical analyses only support $l_\infty$ perturbations.

5. The presentation of this manuscript is relatively poor. For example, (1) the authors should correctly use \citep and \citet to include references; (2) In Equation (1), the relationship between $A^L$, $A^U$ and $A$ is unclear, does this indicate $A^L = A^U$?; (3) You use $Loss_{slope}$ in Equation (5) and $loss_{Slope}$ in Equation (6), please pay attention to the terminology consistency.

Minor:

1. In addition to the linearizing the loss function and randomized smoothing, there are several other types of certified robustness methods, including (1) using geometry: "Provable robustness of relu networks via maximization of linear regions." (2019), "Training Provably Robust Models by Polyhedral Envelope Regularization" (2023) (2) Lipschitz-aware training: "Lipschitz-Margin Training: Scalable Certification of Perturbation Invariance for Deep Neural Networks" (2018)

**Questions:**

Based on the concerns above, I think the current manuscript needs major editing before being considered for publication at a top-iter conference. Please consider the following questions:

1. Discuss about the role of several hyper-parameter, how to set them and conduct ablation studies.

2. Include more baselines and apply the proposed method on that to more comprehensively validate the performance.

3. Provide more justifications about the theoretical analysis in the proof of Lemma 1. Answer the third point in the weakness part.

4. Include the writing to make the manuscript nicer and the terminology consistent.

In addition, I have the following question:

5. When considering the effect of one particular neuron on the local Lipschitz constant, why do you use the criteria in Section 4.1 instead of directly considering the difference between the local Lipschitz if we graft this unstable neuron and the original local Lipschitz. I think this value (the change of local Lipschitz) is more straightforward and not difficult to compute or estimate. In addition, this will facilitate us to select neuron across different layers, as the current criteria in Section 4.1 have different numerical meanings for different layers.

6. Instead of considering the largest sensitivity and the number of unstable inputs in Equation (3) and (4), why don't we consider the average sensitivity among the training set for neuron selection? I believe this will decrease the number of hyper-parameters.

7. It would be better to give a formal definition of "local Lipschitz constant" in the task we consider in this work.

---

> ### Author Response · Authors · 2025-11-25
>
> We sincerely appreciate the reviewer for carefully reading our paper and providing valuable reviews. We update the manuscript as you suggested. In particular, we improve the clarity of the proofs, including the section highlighted by the reviewers. We also incorporate additional baselines, FGSM variants and other certified training to validate the effectiveness of our method. Through an extensive ablation studies, we aim to justify our hyperparameter choices and demonstrate their effects. Due to the character limits, we mention table numbers to point some experiments.
> - Since alpha-beta CROWN is a verifier rather than a training framework, it cannot be directly used for learning. Instead, we add CTBench (ICML 2025) [1], which has demonstrated state-of-the-art performance in recent certified training literature. CTBench employs regular CE, adversarial CE, and verfied CE loss altogether.
> - We follow Theorem 3.2 in [2], where the equality holds when $\epsilon$ = 0 or there are no approximation errors (exactly bounded). If there are no approximation errors, this means that no unstable neurons remain, and the neural network can be represented as a linear network.
>
> [1] Mao, Yuhao, Stefan Balauca, and Martin Vechev. "Ctbench: A library and benchmark for certified training." ICML 2025.
>
> [2] Zhang, Huan, et al. "Efficient neural network robustness certification with general activation functions." NeurIPS 2018.
>
> **A1**. We conduct ablation studies to justify our hyper-parameter choices and the results are presented in Table 12 and 13 in Appendix J of the revised version. Regarding the learning rate of slope loss \beta\, we observe that as \beta\ decreases, SA and RA increase while VA decreases. We believe that this hyper-parameter plays a key role in balancing the trade-off between SA/RA and VA. We also find that overall performances remain similar for $k$ < 6. In addition, we observe that the method exhibits consistent performance regardless of whether neurons are selected based on instability and weighted interval scores around 80% and 15%, respectively.
>
>
> **A2**. We include more baselines, FGSM-MEP [3] and FGSM-PCO [4] to more comprehensively validate the performance of our method. Linearity grafting approach (both Ours and LG) significantly improve their certified robustness. Ours achieves stronger certified robustness (VA), which is generally difficult to improve, while LG still show better SA and RA performances. We believe this occurs because LG tends to select neurons with small activation magnitude, which correspond to very low scores from the weighted interval score. As a result, LG grafts neurons that contribute less to overall sensitivity, leading to better SA/RA but weaker improvements in certified robustness.
>
> [3] Jia, Xiaojun, et al. "Prior-guided adversarial initialization for fast adversarial training." ECCV 2022.
>
> [4] Wang, Zhaoxin, et al. "Preventing catastrophic overfitting in fast adversarial training: A bi-level optimization perspective." ECCV 2024.
>
> **A3**. We noticed that the reduction in approximation error from grafting was not sufficiently explained in the original version. To deal with this, we revise the proof to more clearly illustrate how grafting contributes to reducing the approximation error.
>
> **A4**. As we mentioned earlier, we have updated our manuscript to improve clarity and readability. In particular, several sections, including the proofs and experiments, have been refined to ensure that the proposed ideas are more coherent.
>
> **A5**. In general, directly computing the $\ell_\infty$ Lipschtiz constant and using it during training is computationally infeasible. As shown in [5], obtaining the exact $\ell_\infty$ Lipschitz constant is time consuming, and current methods can handle only very tiny networks. In contrast, the $\ell_2$ Lipschitz constant can be computed and incorporated into the training process. Since our goal is to improve robustness under $\ell_\infty$ perturbations, we did not adopt $\ell_2$ Lipschtiz constant.
>
> [5] Shi, Zhouxing, et al. "Efficiently computing local lipschitz constants of neural networks via bound propagation." NeurIPS 2022.
>
> **A6**. We conduct an additional experiment using average sensitivity and include the result in Table 11 of the revised version. The result shows that average sensitivity leads to better SA but poor VA performance. Similar to the answer A5, we prioritize certified robustness, identifying neurons with the largest sensitivity is more appropriate for our objective.  Moreover, the average sensitivity criterion results in a looser $\ell_\infty$ Lipscthiz constant, which is undesirable for our expectation.
>
> **A7**. We have already included the formal definition of local Lipschitz constant in Section 2.2 of the paper. The local Lipschitz constant describes the rate of change with respect to the $\ell_\infty$ norm, as defined in the manuscript. As mentioned earlier, our focus is not on the $\ell_2$ norm.

---

### Author Response · Authors · 2025-12-04

We sincerely appreciate the ACs and staff for their prompt and thoughtful efforts in addressing the unintended breach of anonymity. We are also deeply grateful to all reviewers for their constructive feedback and insightful comments on our work.

Our work contributes new theoretical insights into Linearity Grafting by analyzing it through the lens of the local $\ell_\infty$ Lipschitz constant. We show that grafting linearity into non-linear activation functions can reduce the approximation error caused by linear relaxation, which is a key bottleneck in verifying neural networks. Even the original LG paper **does not explicitly interpret grafting in terms of approximation error, but rather focuses on reducing the number of unstable neurons.**

Building on this insight, we propose Lipschitz-aware Network Grafting, which achieves tighter local Lipschitz constants and, consequently, stronger certified robustness, **even without certified training**, as confirmed by our experimental results and theoretical proofs.

We believe that adversarial training, when combined with our method, can achieve certified robustness comparable to that of certified training, even though certified robustness is typically difficult to attain through adversarial training alone.

Our contributions are summarized as follows:
- We provide refined theorem and lemmas illustrating that linearity grafting tightens the $\ell_\infty$ Lipschitz constant. Moreover, we show that grafting linearity into the most sensitive neurons leads to a tighter $\ell_\infty$ Lipschitz constant than grafting into the least sensitive neurons.
- We demonstrate that applying our method to existing baselines (FGSM-Align, FGSM-MEP, FGSM-PCO) significantly enhances certified robustness, despite not relying on any certified training.
- Our extensive experiments (large-scale dataset CIFAR-100, other activation functions) and hyper-parameter ablation study show that our method consistently improves certified robustness and outperforms existing approach.
- We also provide an explanation of the trade-off between SA/RA and VA in Limitations. Briefly, under the min–max optimization, certified robustness corresponds to tightening the upper bound on the maximum objective function, while adversarial robustness is driven by optimizing its lower bound. As a consequence, improving certified robustness typically leads to a reduction in SA and RA.
- We would like to clarify that the $\ell_\infty$ Lipschitz constant is so difficult to compute, making it impractical to use directly during training.

To the best of our knowledge, **no prior work** has provided such theoretical insights into linearity grafting. We believe our method offers a meaningful step toward improving certified robustness in a computationally efficient manner, and we look forward to refining it further.

Once again, we sincerely appreciate your efforts.

The authors.

---

### Meta-Review · Area_Chair_Pwse · 2026-01-05

**Summary:**

The reviewers acknowledge the interest of the problem tackled in this work,  However, they express concerns regarding the number of hyper-parameters, the limited baselines used for comparison, some mathematical derivations, the limited scale of the experiments, and the clarity of several aspects of the paper.

**Reviewer Concerns:**

The authors provided comparisons to some additional baselines and empirical results (e.g., on CIFAR-100). Aspects related to the clarity of the presentation and of the mathematical derivations, the hyper-parameters, and other concerns were discussed in the rebuttal but are more subjective. Considering the initial scores and the number of concerns raised by the reviewers, the AC believes that this paper cannot be accepted in its current form and should be revised before resubmission to an upcoming venue.

**Reviewer Scores:**

Although the authors' feedback may have convinced some of the reviewers to slightly raise their scores, it is unlikely that it would have fully convinced them to the point of making this paper pass the ICLR acceptance threshold.

---

### Decision · Program_Chairs · 2026-01-26

Reject